# Gene expression-based identification of prognostic markers in lung adenocarcinoma

**Annette Salomonsson[1], Daniel Ehinger[1,2], Mats Jönsson[1], Johan Botling[3,4], Patrick Micke[4], Hans Brunnström**  [5,6], **Johan Staaf[1,7], Maria Planck[1,8,9]\***

**1** Department of Clinical Sciences Lund, Division of Oncology, Lund University, Lund, Sweden,
**2** Department of Genetics, Pathology, and Molecular Diagnostics, Skåne University Hospital, Helsingborg, Sweden, **3** Department of Laboratory Medicine, Institute of Biomedicine, Sahlgrenska Academy at University of Gothenburg, Gothenburg, Sweden, **4** Department of Immunology, Genetics, and Pathology, Uppsala University, Uppsala, Sweden, **5** Department of Clinical Sciences Lund, Division of Pathology, Lund University, Lund, Sweden, **6** Department of Genetics, Pathology, and Molecular Diagnostics, Skåne University Hospital, Lund, Sweden, **7** Department of Laboratory Medicine, Division of Translational Cancer Research, Lund University, Lund, Sweden, **8** Department of Clinical Sciences Lund, Division of Respiratory Medicine, Allergology, and Palliative Medicine, Lund University, Lund, Sweden, **9** Department of Respiratory Medicine and Allergology, Lund, Sweden

\* maria.planck@med.lu.se

## Abstract

### Introduction

Many studies have aimed at identifying additional prognostic tools to guide treatment choices and patient surveillance in lung cancer by assessing the expression of individual proteins through immunohistochemistry (IHC) or, more recently, through gene expression-based signatures. As a proof-of-concept, we used a multi-cohort, gene expression-based discovery and validation strategy to identify genes with prognostic potential in lung adenocarcinoma. The clinical applicability of this strategy was further assessed by evaluating a selection of the markers by IHC.

### Materials and methods

Publicly available gene expression data sets from six microarray-based studies were divided into four discovery and two validation data sets. First, genes associated with overall survival (OS) in all four discovery data sets were identified. The prognostic potential of each identified gene was then assessed in the two validation data sets, and genes associated with OS in both data sets were considered as potential prognostic markers. Finally, IHC for selected potential prognostic markers was performed in two independent and clinically well-characterized lung cancer cohorts.

### Results and conclusions

The gene expression-based strategy identified 19 genes with correlation to OS in all six data sets. Out of these genes, we selected *Ki67, MCM4* and *TYMS* for further

**Data availability statement:** All relevant data are within the manuscript and its Supporting Information files.

**Funding:** This study was supported by The Swedish Cancer Society (MP), The Sjöberg Foundation (MP), Berta Kamprad´s Cancer Foundation,  Hain's Foundation (MP), The King Gustaf V Jubilee Fund (MP), The Thelma Zoégas Foundation for Medicinal Research (DE), The Stig and Ragna Gorthon Foundation (DE), and the grant from the Swedish state under the agreement between the Swedish government and the county councils, ALF (MP). The funders had no role in the study design, data collection and analysis, decision to publish, or preparation of the manuscript.

**Competing interests:** The authors have declared that no competing interests exist.

assessment with IHC. Although an independent prognostic ability of the selected markers could not be confirmed by IHC, this proof-of-concept study demonstrates that by employing a gene expression-based discovery and validation strategy, potential prognostic markers can be identified and further assessed by a technique universally applicable in the clinical practice. The concept of studying potential prognostic markers through gene expression-based strategies, with a subsequent evaluation of the clinical utility, warrants further exploration.

## Introduction

Despite recent advancements in the understanding and treatment of lung cancer, the prognosis is poor and lung cancer continues to be the leading cause of cancer-related mortality worldwide [1]. Non-small cell lung cancer (NSCLC) accounts for the majority of cases, with adenocarcinoma (AC) as the most frequent histological subtype [2]. Disease stage and patient's performance status are the most well-established and clinically used prognostic factors. Patients with localized disease can be candidates for curatively intended surgery. However, also among these patients, there is a substantial mortality and a 5-year survival rate of only around 60% [3]. For patients with tumors of TNM stage 1B or higher, post-operative adjuvant chemotherapy leads to a decreased risk of recurrence and improved survival [3]. Since recently, the addition of targeted therapy (for *EGFR*-mutated cases) or immunotherapy (for *EGFR*- and *ALK*-negative tumors of stage 2 or higher that show high expression of PDL1), is also recommended [4,5]. The varied outcome for surgically treated patients, also within the same disease stage, illustrates a need for additional tools to guide treatment choices and patient surveillance. With the emergence of yet more strategies involving immunotherapy or targeted therapy in the preoperative and/or postoperative curative setting, treatment decisions will become more and more complex [6–8]. Many studies have aimed at identifying prognostic markers, often by assessing the expression of individual proteins through immunohistochemistry (IHC). However, despite a plethora of IHC studies in lung cancer, no such markers are in clinical use today. More recently, gene expression-based lung cancer signatures turned out as promising prognosticators that deserve further validation for patient benefit in clinical praxis, but the feasibility of such costly and labor-intensive analyses in a clinical routine remain disputable [9]. In this proof-of-concept study, we hypothesized that by utilizing a multi-cohort, gene expression-based discovery and validation strategy, we could identify genes with prognostic potential in lung adenocarcinoma. Subsequently, to increase a potential clinical applicability of this strategy for identifying prognostic markers, a selection of the identified markers was further assessed by IHC.

## Materials and methods

All analytical steps, and the public and in-house lung cancer cohorts that we used, are outlined in Fig 1. In brief, we explored six different publicly available gene

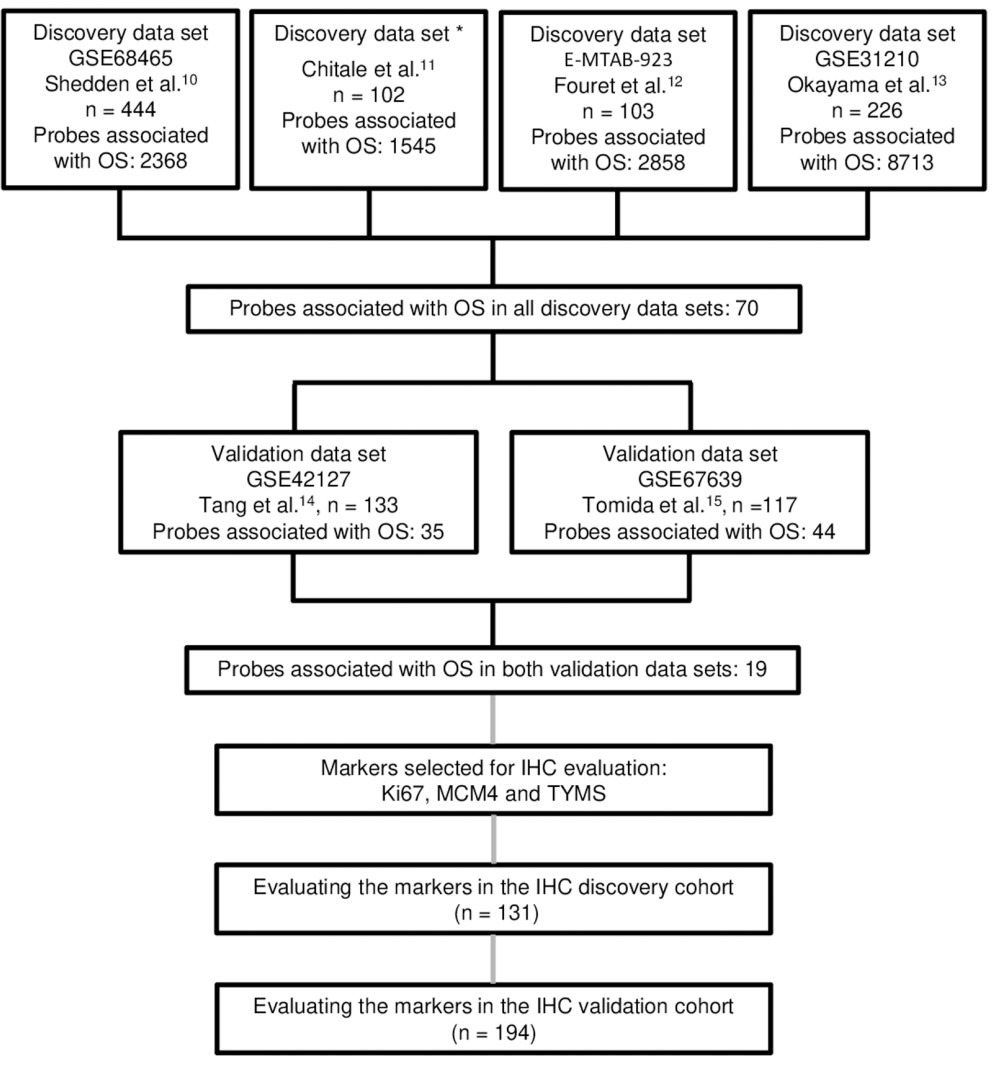

**Fig 1. Schematic image of the gene expression-based strategy for identification of prognostic markers and subsequent IHC evaluation.** For each probe (matching to a gene) in the four discovery data sets, the median gene expression value was used to divide the samples into two groups (high/low). The log-rank test was employed to identify probes significantly associated with OS (P-value < 0.05). Results from the four discovery data sets were then compared and probes that were significantly associated with OS in all four data sets were tested in the same manner in two validation data sets. The genes significantly associated with OS in both data sets were classified as potential prognostic markers. Out of these genes, three were selected for IHC evaluation in two patient cohorts. One of the cohorts was used as an IHC discovery cohort were optimal cut-offs for each markers were selected. These cut-offs were then applied to the cases in the IHC validation cohort. * Only the U133 2plus array subset (n = 102) from Chitale et al. was included. Abbreviations: OS = overall survival, IHC = immunohistochemistry.

expression data sets, in total comprising 1,125 lung adenocarcinomas, to identify and validate markers consistently associated with overall survival (OS) and then evaluated a selection of these markers by IHC in two independent cohorts.

## Gene expression-based discovery and validation

Publicly available transcriptomic profiles and matched survival data were obtained from six microarray-based lung cancer studies [10–15]. Samples with AC histology (n = 1,125) were chosen for further comparisons of the gene expression data, which were processed as previously described [16]. Four of the data sets were used in the discovery step, all based on

the Affymetrix platform [10–13]. For each probe (matching to a gene) in the data sets, the median gene expression value was used to divide the samples into two groups (high/low). Then, the log-rank test was employed to identify probes significantly associated with OS ($P$-value < 0.05). Results from the four discovery data sets were then compared and probes that were significantly associated with OS in all four data sets advanced to the validation step, in which the probes generated from the discovery step were tested in the same manner in two validation data sets, based on non-Affymetrix platforms [14–15]. The genes significantly associated with OS in both data sets were then classified as potential prognostic markers.

## Immunohistochemical evaluation of potential prognosticators

Among the potential prognostic markers obtained by our discovery and validation strategy, we selected three genes (*Ki67*, *MCM4* and *TYMS,* as further discussed below) for further IHC evaluation of the corresponding proteins. Immunohistochemical staining was performed in two independent and clinically well-characterized lung cancer cohorts. The first cohort was used as an IHC discovery cohort for identification of cut-offs for classifying samples as having a low or high expression of each marker, and the second cohort was used as an IHC validation cohort, where these cut-offs were then applied.

The IHC discovery cohort was based on the "Southern Swedish Lung Cancer Study", which, prospectively and after written informed consent, included patients with primary lung cancer who underwent surgical treatment at the Skåne University Hospital, Lund, Sweden, between the first of January 2005 and the last of December 2011 [17]. The present investigation included 131 AC from the Southern Swedish Lung Cancer Study. The IHC validation cohort was based on 194 AC cases from the "Uppsala NSCLC II cohort" which included patients with primary lung cancer who underwent surgical treatment at the University Hospital in Uppsala, Sweden, between the first of January 2006 and the last of December 2010 [18,19]. Patient characteristics and clinicopathological data were described previously [20]. The two IHC cohorts were balanced regarding age, gender, smoking, stage, and treatment. The studies were approved by the Regional Ethical Review Board in Lund (Dnr 2004/762) and Uppsala (Dnr 2012/532) and conducted in adherence with the Declaration of Helsinki.

Only patients that were surgically treated for primary NSCLC tumors were included. Patients receiving neoadjuvant treatment, or chemotherapy for another malignancy six months before surgery, were excluded from the present study. All cases were previously reviewed by two pathologists (HB and PM), who updated the diagnoses in accordance with the 2015 WHO classification and TNM 7 and who confirmed all changes from the original diagnoses [17,19,21,22]. Furthermore, growth patterns were evaluated (HB) for stratification into three groups: minimally invasive/predominant lepidic, predominant acinary/papillary, and mucinous or predominant micropapillary/solid. Overall survival data were retrieved from the Swedish Cancer Registry, to which reporting is mandatory by law. The registry was consulted on June 26, 2018 (the IHC discovery cohort), and on March 29, 2019 (the IHC validation cohort). For one patient in the IHC validation cohort, survival data were unavailable. Analysis of recurrence-free interval (RFI) was performed as previously described [20], and included 122 AC in the IHC discovery cohort, and 164 AC in the IHC validation cohort. Tissue microarrays (TMA) were used for IHC analysis. The TMA-blocks had, for each case, three (the IHC discovery cohort) or two (the IHC validation cohort) cores, 1 mm in diameter. For IHC analysis, 4-µm thick sections were stained according to S1 Table. The slides were scanned and evaluated using the pathXL software (Philips, Amsterdam, The Netherlands). For further analysis, we required a minimum of 200 assessable tumor cells on the TMAs, with most cases having over 1000 evaluable cells.

All stainings were evaluated by three independent observers (AS, DE and MJ) who were blinded to clinical data and patient outcome. Nuclear staining for Ki67 and MCM4 was considered positive. Cytoplasmic or nuclear staining were considered positive for TYMS, though only cells with visible nuclei were counted. Attention was paid to exclude stained non-tumor cells. In case of varying expression of the marker between the cores within a sample, the mean proportion of cells expressing the marker across all cores was assessed. For cases with differences in the scoring between the evaluators, the cases were jointly reviewed, and consensus was reached.

In the IHC discovery cohort, the fraction of viable tumor cells expressing the marker was scored as 0 (0–1%), 1 (>1–10%), 2 (>10–25%), 3 (>25–50%), 4 (>50–75%) or 5 (>75%). For TYMS, in addition to the recorded fraction of positive tumor cells, the staining intensity was scored as 0 (negative), 1 (mild), 2 (moderate), or 3 (strong) and a final score was constructed by multiplying these two parameters thus ranging from 0–15 points. To establish the optimal cut-off for each marker for categorizing samples into high or low expression groups, Kaplan-Meier plots with log-rank tests were used and the cut-offs yielding the lowest p-values in the log-rank tests were selected..

In the IHC validation cohort, tumors were scored as high or low expressors using the optimized cut-offs selected in the IHC discovery cohort. Furthermore, in both cohorts, the combined prognostic ability of the three markers was examined by each case receiving one point per positive maker, thus resulting in a combined score ranging from 0 to 3 points.

### Gene expression of *Ki67*, *MCM4* and *TYMS* in the IHC validation cohort

Gene expression data of *Ki67*, *MCM4,* and *TYMS* were available for 104 AC cases in the IHC validation cohort. Gene expression data are available as GSE81089 and RNA sequencing analysis was performed as previously described by Djureinovic et al. [23]. Samples were classified as having low or high gene expression levels of *Ki67*, *MCM4* or *TYMS* by dividing the samples into two equally sized groups.

### Statistical analysis

Kaplan-Meier plots with log-rank test were used for OS analyses and for analyses of RFI. Univariable and multivariable Cox proportional hazards regression models were used for further comparisons between groups. Multivariable models were adjusted for stage (I, II, III, and IV), age, smoking status (current, past, or never), gender, adjuvant therapy, growth pattern, and patients' performance status (the latter available for the IHC validation cohort only). Spearman's rank correlation was used to assess the correlations between gene expression levels of the potential prognostic genes. The Mann-Whitney U test/Wilcoxon rank-sum test and Fisher's exact test were used to compare data between groups. A *P*-value < 0.05 was considered statistically significant. All statistical analyses were performed using R (version 3.6.1) [24].

## Results

### Gene expression-based identification of genes with prognostic potential

For 70 probes (genes), the gene expression levels were associated with OS in all four discovery data sets, as schematically illustrated in Fig 1. Of these, 19 genes (listed in Table 1) were associated with OS in the two gene expression data sets used for validation and were thus considered as having prognostic potential in lung adenocarcinoma.

In the two validation data sets, correlation plots (Fig 2) showed a generally strong correlation in gene expression levels between the 19 genes. Broadly, the 19 genes could be divided into two groups that were inversely correlated to each other. By using Kaplan-Meier plots, it was demonstrated that high gene expression levels were associated with worse outcome for one group of genes, while low expression levels were associated with worse outcome for the other group, as exemplified in S1 Fig.

### Evaluation of the clinical utility

To further explore the potential clinical utility of our gene-expression based strategy for identifying prognostic markers, a selection of the identified markers was further assessed by IHC. Considering clinical practicability, we preferred markers where high expression was associated with worse outcome. Also, availability of reliable antibodies was considered in selecting candidate genes for further analyses with IHC. In the two validation data sets, the patients classified as having high gene expression levels of *Ki67*, *MCM4*, and *TYMS* did not fully overlap (S2 Fig), thereby suggesting that the three markers could possibly complement each other. Based on these considerations, we chose these three genes for further

**Table 1. Genes with prognostic potential identified in the gene expression-based discovery and validation step.**

| Gene Symbol | Gene Name |
| --- | --- |
| KI67 | Marker of proliferation Kiel 67 |
| MCM4 | Minichromosome maintenance complex component 4 |
| TYMS | Thymidylate synthetase |
| CCNA2 | Cyclin A2 |
| CCNE1 | Cyclin E1 |
| BUB1B | Budding uninhibited by benzimidazoles 1 homolog beta |
| DLGAP5 | Discs large homolog associated protein 5 |
| KIF14 | Kinesin family member 14 |
| NUSAP1 | Nucleolar and spindle-associated protein 1 |
| RACGAP1 | Rac GTPase activating protein 1 |
| ECT2 | Epithelial cell transforming sequence 2 oncogene |
| ASPM | Abnormal spindle-like microcephaly-associated protein |
| PRC1 | Protein regulator of cytokinesis 1 |
| BTG2 | B-cell translocation gene 2 |
| HLF | Hepatic leukemia factor |
| GDF10 | Growth differentiation factor 10 |
| CTTN | Cortactin |
| COL4A3 | Collagen, type IV, alpha 3 |
| CIRBP | Cold inducible RNA binding protein |

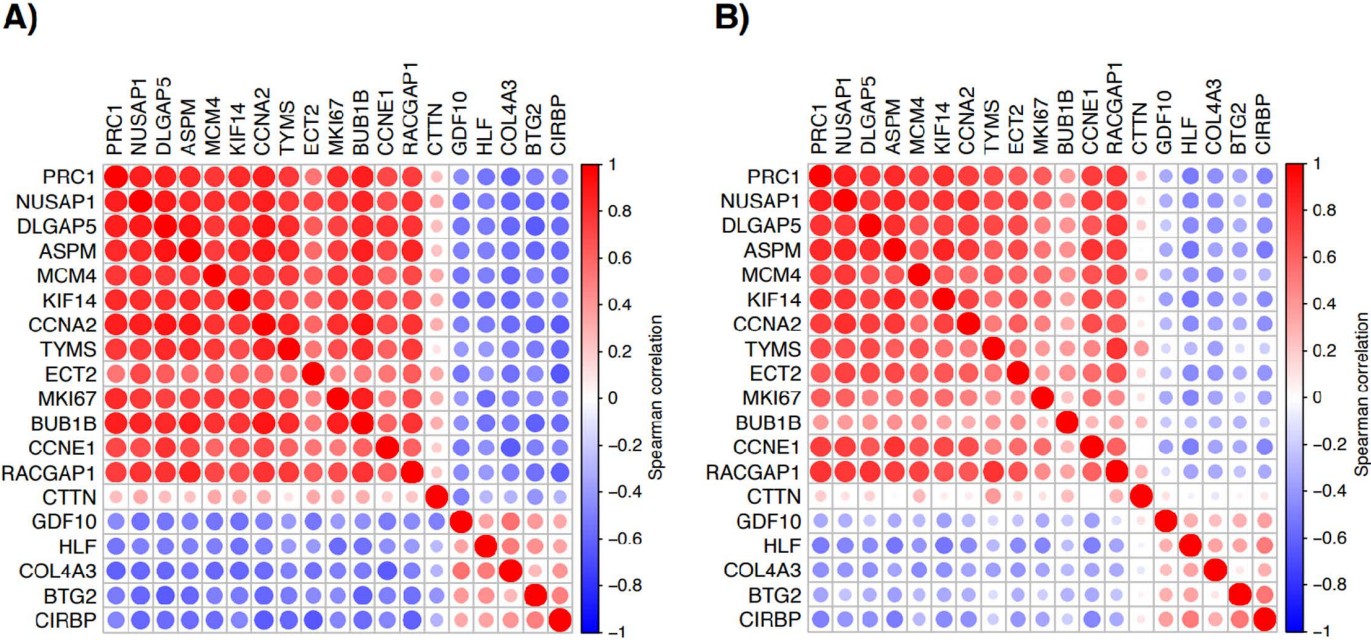

**Fig 2. Spearman correlation of gene expression levels of the 19 candidate genes in the two validation data sets.** (A) Tomida et al. [15], (B) Tang et al [14]. If multiple probes were available for a gene then the probes with largest standard deviation was chosen to represent the gene. Area of the circles show the absolute value of corresponding correlation coefficients.

analyses with IHC in two independent lung cancer cohorts. Representative microscopic images of the stainings for Ki67, MCM4 and TYMS are shown in S3 Fig.

## Protein expression of Ki67, MCM4 and TYMS in the IHC discovery cohort

For AC in the IHC discovery cohort, the protein expression could be evaluated for Ki67 in all 131 cases, for MCM4 in 129 cases and for TYMS in 120 cases. For Ki67, a cut-off of > 10% positive tumor cells most clearly identified prognostic groups in the OS analysis (log rank p-value 0.014) among the AC cases and was therefore chosen for identification of samples with a low or high expression (Fig 3A). By applying this cut-off, 74 AC cases (56%) were classified as having a high Ki67 protein expression. A cut-off of > 75% positive tumor cells was selected for MCM4 among the AC cases in the OS analysis (log rank p-value 0.0005), which resulted in 15 cases (12%) identified as having a high MCM4 expression (Fig 3B). For TYMS, a score (obtained by multiplying fraction and intensity, as further explained in Materials and Methods) of > 2 points was chosen for identification of AC samples with a high TYMS expression in the OS analysis, which resulted in 19 cases (16%) classified as having a high expression of TYMS (Fig 3C), but was not statistically significant in the OS analysis (log rank p-value 0.055). The prognostic value of these cut-offs for Ki67, MCM4, and TYMS in the RFI-analysis are shown in S4 Fig.

For the selected cut-offs in the IHC discovery cohort, smoking status was significantly associated with Ki67 expression levels, with low expression often observed in never smokers and high expression in current smokers (Fisher's exact test, $P = 0.007$). For MCM4 and TYMS, no associations with smoking were found. Furthermore, expression of Ki67 was associated with AC growth pattern, as samples with a high expression were more frequently found in the group with mucinous or predominant micropapillary/solid pattern (Fisher's test, $P = 0.03$). For MCM4 and TYMS, no associations with growth patterns were found. For age, gender, stage, and number of cases receiving adjuvant treatment, no associations between these parameters and patients with a high or low expression of Ki67, MCM4 or TYMS, respectively, were found.

## Protein expression of Ki67, MCM4 and TYMS in the IHC validation cohort

In the IHC validation cohort, the protein expression of Ki67, MCM4, and TYMS could be assessed in 159, 178, and 146 AC cases, respectively. By applying the identified cut-offs from the IHC discovery cohort for respective gene, high expression was found in 91 cases (57%) for Ki67, in 17 cases (10%) for MCM4, in and 17 cases (12%) for TYMS. High expression of Ki67, MCM4, and TYMS was associated with male gender (Fisher's test, $P < 0.05$ in all three tests) and high expression of Ki67 was associated with more advanced stages (stage III) (Fisher's test, $P = 0.007$). Furthermore, the expression of Ki67 and MCM4 was associated with growth pattern, as proportionally more cases with a high expression were found in the group with mucinous or predominant micropapillary/solid pattern compared to cases with a low expression, where proportionally more cases were minimally invasive/lepidic or acinary/papillary (Fisher's test, $P < 0.05$). The expression of Ki67 was associated with smoking as there were proportionally more never smokers among cases with a low expression compared to cases with a high expression (Fisher's test, $P < 0.001$). Apart from these findings, no other associations between age, gender, stage, smoking status, growth pattern, WHO performance status, and number of cases receiving adjuvant treatment and patients with a high or low expression of Ki67, MCM4 or TYMS, respectively, were found.

High protein expression of Ki67 was associated with a worse prognosis in the 5-year OS analysis (log-rank test, $P = 0.0002$, Fig 4A). In the univariable Cox proportional hazards regression model, Ki67 expression was significantly associated with prognosis (HR 2.54, 95% CI 1.54–4.21). However, these results did not remain statistically significant in the multivariable model adjusted for stage, growth pattern, age, gender, smoking, WHO performance status, and adjuvant treatment (HR 1.29, 95% CI 0.68–2.45, S2 Table). In the RFI analysis, patients with a high expression of Ki67 had a higher rate of recurrence (log-rank test, $P = 0.0003$, S5A Fig). For MCM4 and TYMS, no statistically significant associations between protein expression and survival or RFI could be demonstrated (Figs 4B, 4C, S5B and S5C).

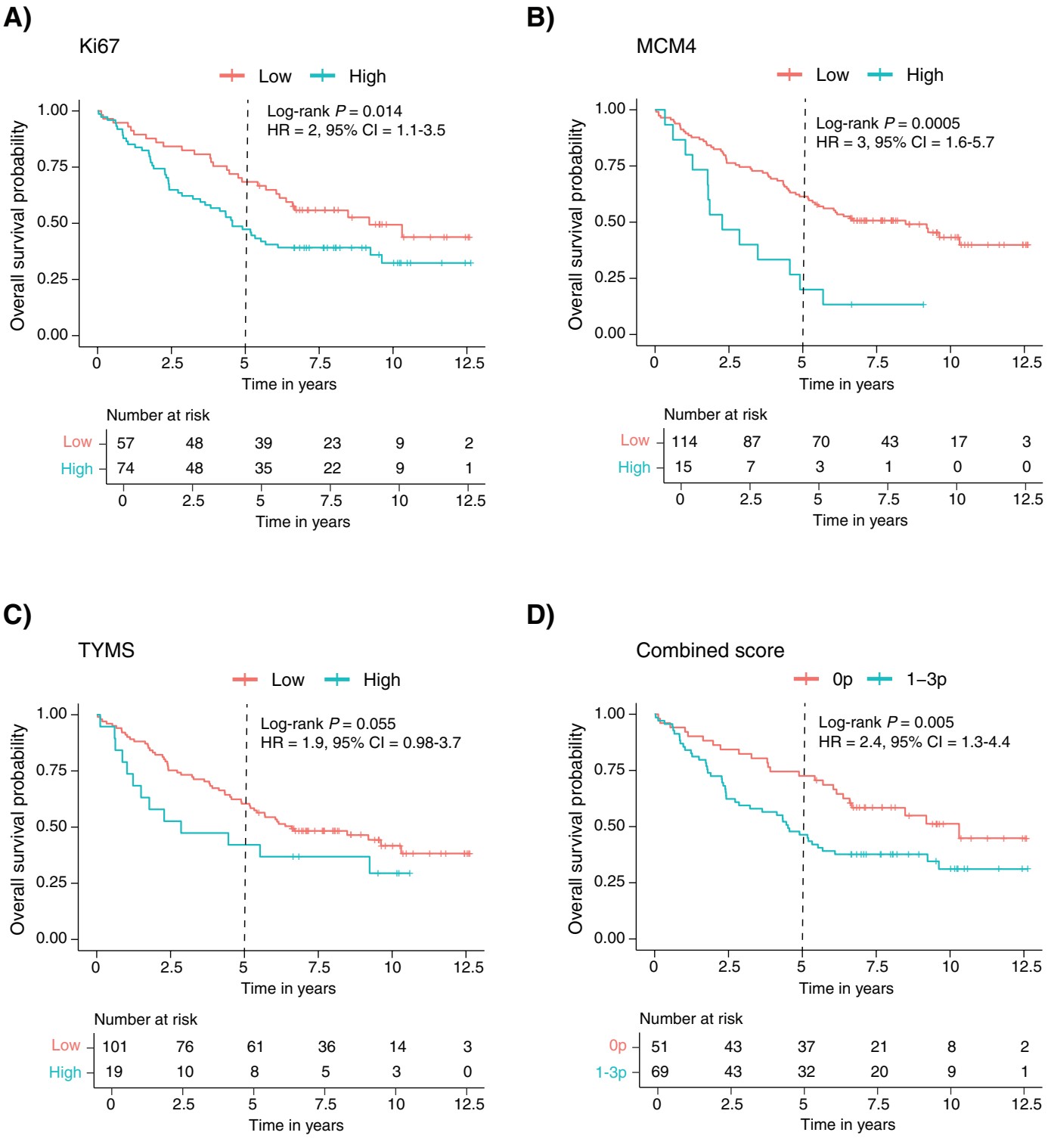

**Fig 3. Prognostic value of Ki67 (A), MCM4 (B), TYMS (C), and combined score (D), on overall survival in the IHC discovery cohort.**

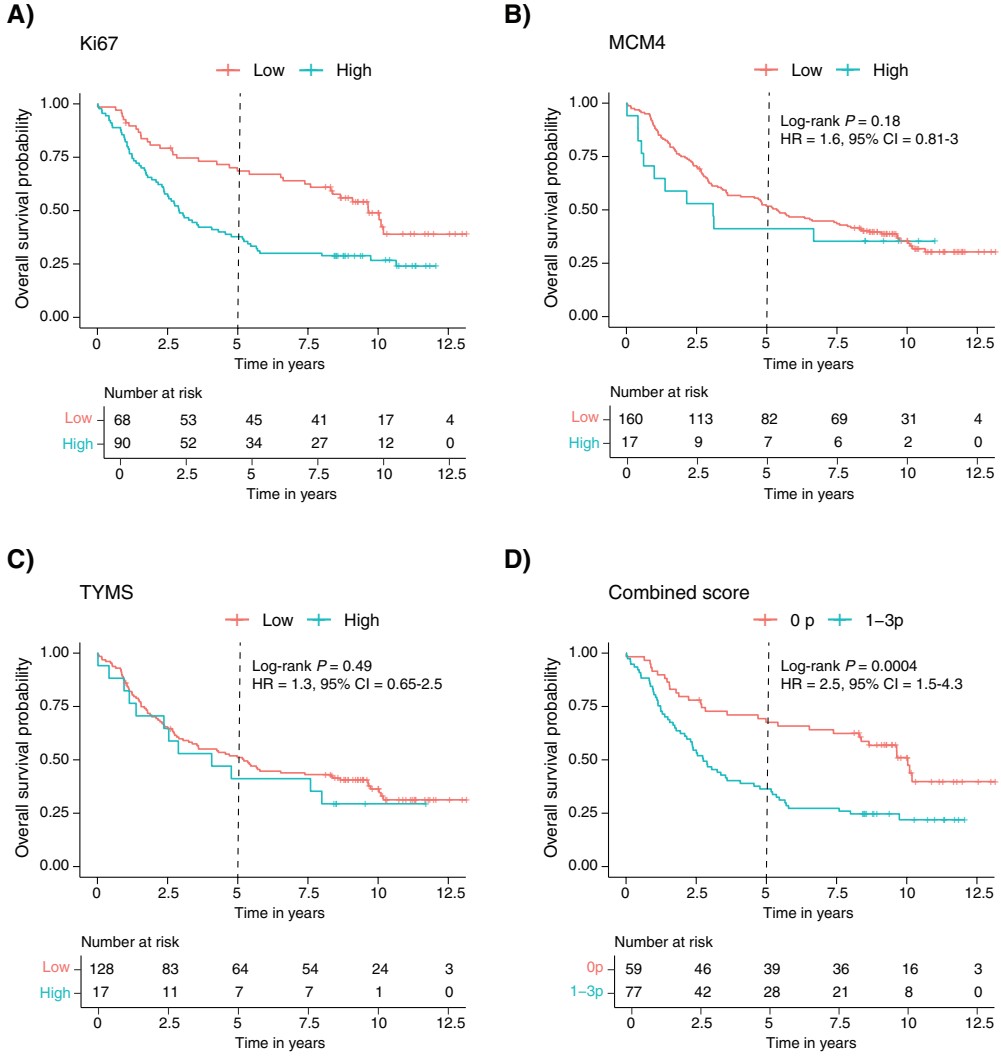

**Fig 4. Prognostic value of Ki67 (A), MCM4 (B), TYMS (C), and combined score (D), on overall survival in the IHC validation cohort.**

## Combining markers for improved prognostication

Considering the gene expression findings, we also examined the combined prognostic ability of the three markers. The number of patients positive for each marker, and the overlap between these, are presented in S6 Fig. All three markers did not independently add prognostic information in the combined score as there were no patients that were positive for only MCM4 in both cohorts, and TYMS only added one (IHC validation cohort) or two (IHC discovery cohort) cases to the high-risk group (more than one point) compared to Ki67 alone.

In both IHC cohorts, cases that were positive for one or more markers had a worse prognosis in the 5-year OS analysis (Figs 3D and 4D) and a higher rate of recurrence (S4D and S5D Figs) compared to cases that were negative for all three markers. However, in the IHC validation cohort, these associations did not remain statistically significant in the multivariate model.

### Gene expression of *Ki67*, *MCM4* and *TYMS* in the IHC validation cohort

Gene expression levels of *Ki67, MCM4*, and *TYMS*, obtained by RNA sequencing analysis, were available for 104 AC cases in the IHC validation cohort. Out of these 104 cases, IHC data were missing for 20 cases for Ki67, four cases for MCM4, and 28 cases for TYMS. For all three markers, an association between gene expression levels and IHC classification (low or high expression) could be observed (Wilcoxon test, $P<0.01$ all three tests). However, and most evident for MCM4 and TYMS, there were also cases with high gene expression that were classified as low expressors by IHC (S7 Fig).

The prognostic value of *Ki67*, *MCM4*, and *TYMS* gene expression levels in the IHC validation cohort were evaluated by dividing the samples into two equally sized groups. For *Ki67* and *TYMS*, no statistically significant differences between the groups could be identified, although potentially prognostic subgroups could be visualized in the Kaplan-Meier plots (S8 Fig). For *MCM4*, patients with high expression levels had a worse prognosis compared to patients with low expression levels in the 5-year OS analysis (log-rank test, $P=0.004$, S8 Fig).

### Discussion

Studies of potential prognosticators in lung cancer are often based on the immunohistochemical expression of protein markers or on gene expression-based prognostic signatures. There are no prognostic IHC markers in clinical use for lung cancer today, and the reproducibility and clinical benefit of gene expression-based prognostic signatures needs to be thoroughly validated before being implemented in a clinical setting [9]. However, gene expression can already now be employed as a research tool for identifying potential prognosticators. As a proof-of-concept, we identified markers with prognostic impact in lung adenocarcinoma through a gene expression-based, multi-cohort discovery and validation strategy. Based on global gene expression profiling array data sets generated from original analyses of early (operable stage I-III) lung adenocarcinomas in North American, European, and Japanese patient populations, we were able to associate the expression of 19 genes to survival in a total of six independent studies, published in reputable journals [10–15]. We also selected markers identified by this strategy for further evaluation by IHC, a method more adapted to the current clinical setting, thus underlining a potential future clinical utility.

Several of the 19 potential prognostic markers that we identified in our expression-based discovery and validation strategy (Table 1) are linked to proliferation, and either higher expression levels (e.g., for *Ki67*) or lower expression levels (e.g., *BTG2*) are associated with poor outcome [25,26]. The prognostic impact of proliferation has long been recognized in many types of cancer, and many IHC-based markers target proliferation [27]. Furthermore, it has been suggested that proliferation-associated genes are key components in gene-expression-derived adenocarcinoma prognostic phenotypes [28]. Accordingly, genes linked to proliferation proved important in our current multi-cohort approach to associate gene expression with patient overall survival. As illustrated in correlation plots for the two validation data sets (Fig 2), the 19 candidate genes could broadly be divided into two groups that were inversely correlated to each other. In the larger of these two groups, all genes are directly implicated in proliferation.

To further explore the potential clinical applicability of our gene-expression based strategy for identifying prognostic markers, we selected three of the 19 markers for further assessment. The gene expression levels of the three selected markers (*Ki67*, *MCM4*, and *TYMS*) were correlated to each other in the two validation data sets, and high expression was associated with worse prognosis. Furthermore, as illustrated in S2 Fig, the markers could possibly complement each other in identifying high-risk patients. Based on the gene expression correlation analyses, it could be hypothesized that alternative gene selections would have resulted in similar results. IHC has the advantage of being an accessible and applicable method in the clinical routine and, for all three markers, there are reliable antibodies available and the genes have a recognized prognostic potential in lung adenocarcinoma [26,29–32]. Difficulties in standardization and reproducibility across IHC studies remain challenging, but may improve with the emergence of digital image analysis [33]. However, other methods suitable for formalin-fixed paraffin embedded tissue, such as RNA-based NanoString technology or quantitative reverse transcription polymerase chain reaction (qRT-PCR), might have also been considered.

The robustness of the three selected markers was demonstrated through a clear correlation between gene expression levels and IHC classification in the IHC validation cohort. Furthermore, when assessing the prognostic value of gene expression levels of the three markers, potentially prognostic subgroups could be visualized in the Kaplan-Meier plots, although not statistically significant for *Ki67* and *TYMS*. For the IHC stainings, we were able to test the consistency of the chosen cut-offs by using two independent lung cancer cohorts. The first cohort was used to establish cut-off values for categorizing samples into groups (high and low expression). Subsequently, these cut-offs were applied when evaluating the cases in the IHC validation cohort. However, based on these cut-offs, we could only confirm the prognostic ability of Ki67, although the association did not remain prognostic in the multivariable model. For MCM4 and TYMS, the cut-offs chosen in the IHC discovery cohort identified only a small proportion of the samples with a worse prognosis. It is possible that a lower cut-off for these two markers, identifying more patients and more resembling the cut-off chosen for Ki67, would have performed better in the validation cohort.

As illustrated in S2 Fig, more high-risk patients could be identified by combining the three markers compared to using one single marker for gene expression levels of *Ki67*, *MCM4* and *TYMS* in the two validation data sets. However, for the IHC evaluations in our study, the combined score was in both cohorts dependent on the prognostic ability of Ki67 alone, and the patients identified by MCM4 and TYMS overlapped with the patients identified by Ki67 (S6 Fig). These results possibly reflect that we set the cut-offs for MCM4 and TYMS at a level where these markers identified a too small proportion of the patients. Also, all three selected markers are associated with proliferation and, as such, may be redundant as they assess the same cancer characteristic. Possibly, a combination of markers that assess different biological processes could better identify additional high-risk patients. However, in a previous study by Grinberg et al., a prognostic model based on a biomarker panel consisting of five protein markers with diverse biological functions was developed, where each marker also was associated with prognosis in gene expression data sets [34]. When the model was applied to the validation cohort, it failed to improve survival prediction beyond clinical parameters alone, thus questioning the prognostic impact of protein biomarkers and further stresses the difficulties of implementing additional prognosticators into clinical practice.

Our study has several limitations. Out of the 19 potential prognostic markers generated in the gene expression-based step, we selected three for further evaluation with IHC in this proof-of-concept study. It is possible that choosing other, or more, markers would have yielded a different result. Ideally, more AC cases in the IHC-cohorts would have permitted more extensive evaluations and subgroup analyses of the prognostic value of the markers. The markers were further evaluated by using TMAs instead of whole tumor sections, which could have an impact on the validity of the results, particularly when assessing markers with unknown intra-tumoral heterogeneity. The cut-off values were determined by using log-rank tests in the IHC discovery cohort, and the threshold with the lowest p-value from these log-rank tests was considered the optimal cut-off which were then applied to the IHC validation cohort. It is conceivable that another method for identifying an optimal cut-off would have resulted in a different, and perhaps better balanced, cut-off for the markers. The lack of consensus in how to set cut-off values highlight some of the challenges with conducting IHC-based prognostic studies.

To summarize, through our gene expression-based discovery and validation strategy, we identified 19 genes with prognostic potential in lung adenocarcinoma and assessed three of these markers further by IHC. In conclusion, this proof-of-concept study demonstrates that a gene-expression based strategy for identifying prognostic markers, combined with a subsequent evaluation of the clinical utility, is a justified approach that warrants further exploration.

## Supporting information

**S1 Table. Immunohistochemical stainings for Ki67, MCM4 and TYMS.**
(DOCX)

**S2 Table. Multivariate Cox regression analysis for 5-year overall survival for Ki67, MCM4, and TYMS protein expression in the IHC validation cohort.**
(DOCX)

**S1 Fig. Prognostic value of *Ki67* (A), *TYMS* (B), *GDF10* (C), and *CIRBP* (D) gene expression levels in one of the validation data set (Tang et al. [14]).**
(PDF)

**S2 Fig. The overlap between cases with high gene expression levels (cut-off based on the median gene expression values for each gene) of *Ki67*, *TYMS*, and *MCM4* in the two validation data sets.** (A) Tomida et al. [15], (B) Tang et al. [14].
(PDF)

**S3 Fig. Representative microscopic images of the stainings.** For Ki67 and MCM4, nuclear staining was considered positive. For TYMS, cytoplasmic or nuclear staining were considered positive, although only cells with visible nuclei were counted. (A) Low expression of Ki67. (B) High expression of Ki67. (C) Low expression of MCM4. (D) High expression of MCM4. (E) Low expression of TYMS. (F) High expression of TYMS.
(DOCX)

**S4 Fig. Prognostic value of Ki67 (A), MCM4 (B), TYMS (C), and combined score (D), on recurrence-free interval (RFI) in the IHC discovery cohort.**
(PDF)

**S5 Fig. Prognostic value of Ki67 (A), MCM4 (B), TYMS (C), and combined score (D), on recurrence-free interval (RFI) in the IHC validation cohort.**
(PDF)

**S6 Fig. The overlap between cases that were positive for the three markers in the IHC discovery cohort (A) and the IHC validation cohort (B).**
(DOCX)

**S7 Fig. The association between gene expression levels (y-axis, logFPKM) and immunohistochemical classification (x-axis, low or high expression) for the three markers in the IHC validation cohort.**
(PDF)

**S8 Fig. The prognostic value of Ki67 (A), MCM4 (B), and TYMS (C) gene expression levels in the IHC validation cohort.**
(PDF)

## Author contributions

**Conceptualization:** Maria Planck, Annette Salomonsson, Johan Staaf.

**Data curation:** Maria Planck, Annette Salomonsson, Daniel Ehinger, Johan Botling, Patrick Micke, Hans Brunnström, Johan Staaf.

**Funding acquisition:** Maria Planck, Daniel Ehinger.

**Investigation:** Maria Planck, Annette Salomonsson, Daniel Ehinger, Mats Jönsson, Hans Brunnström, Johan Staaf.

**Methodology:** Daniel Ehinger, Maria Planck, Mats Jönsson, Johan Staaf.

**Project administration:** Maria Planck.

**Resources:** Maria Planck, Johan Botling, Patrick Micke, Hans Brunnström, Johan Staaf.

**Supervision:** Maria Planck, Hans Brunnström, Johan Staaf.

**Writing – original draft:** Annette Salomonsson, Daniel Ehinger.

**Writing – review & editing:** Maria Planck, Mats Jönsson, Johan Botling, Patrick Micke, Hans Brunnström, Johan Staaf.

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
