## [Decision Letter · Decision Letter 0]

20 Nov 2024

PONE-D-24-37221Gene expression-based identification of prognostic markers in lung adenocarcinomaPLOS ONE

Dear Dr. Planck,

Thank you for submitting your manuscript to PLOS ONE. After careful consideration, we feel that it has merit but does not fully meet PLOS ONE’s publication criteria as it currently stands. Therefore, we invite you to submit a revised version of the manuscript that addresses the points raised during the review process. The paper is reviewed by two experts in the field and their comments are appended at the end of the letter. Please address all comments. In addition to the reviewer comment please include a data regarding validation of antibodies used in this study.

We look forward to receiving your revised manuscript.

Kind regards,

Asmerom Tesfamariam Sengal, MD, PhD

Academic Editor

PLOS ONE

Journal Requirements:

3. Thank you for stating the following in your Competing Interests section: [The authors declare that they have no competing interests.]. Please complete your Competing Interests on the online submission form to state any Competing Interests. If you have no competing interests, please state "The authors have declared that no competing interests exist.", as detailed online in our guide for authors at http://journals.plos.org/plosone/s/submit-now This information should be included in your cover letter; we will change the online submission form on your behalf.

4. We note that your Data Availability Statement is currently as follows: [All relevant data are within the manuscript and its Supporting Information files.] Please confirm at this time whether or not your submission contains all raw data required to replicate the results of your study. Authors must share the “minimal data set” for their submission. PLOS defines the minimal data set to consist of the data required to replicate all study findings reported in the article, as well as related metadata and methods (https://journals.plos.org/plosone/s/data-availability#loc-minimal-data-set-definition). For example, authors should submit the following data: - The values behind the means, standard deviations and other measures reported; - The values used to build graphs; - The points extracted from images for analysis. Authors do not need to submit their entire data set if only a portion of the data was used in the reported study. If your submission does not contain these data, please either upload them as Supporting Information files or deposit them to a stable, public repository and provide us with the relevant URLs, DOIs, or accession numbers. For a list of recommended repositories, please see https://journals.plos.org/plosone/s/recommended-repositories. If there are ethical or legal restrictions on sharing a de-identified data set, please explain them in detail (e.g., data contain potentially sensitive information, data are owned by a third-party organization, etc.) and who has imposed them (e.g., an ethics committee). Please also provide contact information for a data access committee, ethics committee, or other institutional body to which data requests may be sent. If data are owned by a third party, please indicate how others may request data access.

5. We notice that your supplementary [Supplementary Table 1 and Supplementary Figures 1-8] are included in the manuscript file. Please remove them and upload them with the file type 'Supporting Information'. Please ensure that each Supporting Information file has a legend listed in the manuscript after the references list.

Reviewers' comments:

Reviewer's Responses to Questions

**Comments to the Author**

1. Is the manuscript technically sound, and do the data support the conclusions?

Reviewer #1: Yes

Reviewer #2: Yes

2. Has the statistical analysis been performed appropriately and rigorously?

Reviewer #1: Yes

Reviewer #2: Yes

3. Have the authors made all data underlying the findings in their manuscript fully available?

Reviewer #1: Yes

Reviewer #2: Yes

4. Is the manuscript presented in an intelligible fashion and written in standard English?

Reviewer #1: Yes

Reviewer #2: Yes

5. Review Comments to the Author

Reviewer #1: The authors use publicly available microarray datasets to identify genes whose expression is associated with OS differences in lung adenocarcinoma and then tests clinical utility of select genes using IHC in Swedish cohorts. While correct statistical analysis was used and all conclusions are supported by the data, I had some minor concerns where further details or clarification are needed.

Reviewer #2: The research paper presents a comprehensive study on non-small cell lung cancer (NSCLC) data sets, where, via the means of finding strong statistical associations between gene expression and cancer patient overall survival data, the authors have extracted a list of genes serving as potential prognostic markers for adenocarcinoma-type lung cancer. These markers’ prognostic ability was then assessed on a protein level via immunohistochemistry (IHC), where protein expression was evaluated and its association with overall survival and recurrence-free interval in two patient cohorts was statistically tested.

Overall, the manuscript is well written and shows a strong methodology for predicting prognostic markers for NSCLC using gene expression data. The statistical analyses applied were appropriate for the research question and tested the data stringently. The negative findings (i.e. the results that did not reach statistical significance) were addressed in the discussion section, and the limitations of the study were acknowledged.

High-quality data sets were used for the discovery of prognostic markers and yielded excellent results, revealing a set of correlated genes involved in tumour proliferation. Further IHC experiments were designed with care and were able to demonstrate the promising potential of gene expression-based prognostic research, confirming some, but not all, of the associations with patient survival on the level of both gene and protein expression. Although the IHC validation data set has revealed few statistically significant results, using different expression level cut-offs, as noted by the authors, could potentially result in more positive findings. It also appears that multivariate Cox proportional hazards regression models were another high-impact factor that lowered the findings’ significance levels. It is possible that the size of the data set was too small relative to the number of predictor variables used, resulting in higher p-values; however, this may be subjective, as no standard cut-off for events per variable (EPV) has been defined in current literature.

With regard to manuscript revisions, I only have suggestions concerning a few minor points. I would recommend the authors to consider substituting the term “correlation” where it is not in the context of the actual correlation analyses. For example, on page 4, lines 31-32: “… to identify and validate markers with consistent correlation to overall survival (OS)”. The term “correlation”, although semantically correct, may be misleading here, as the relationship between OS and marker gene expression was established via a log-rank test. It would be better to replace the term “correlation” with “association” as it was already done in several parts of the text and within Figure 1.

Several minor typos and grammatical errors were spotted, therefore additional proofreading is recommended.

After these minor revisions are made, I recommend this manuscript for publication.

6. PLOS authors have the option to publish the peer review history of their article (what does this mean? ). If published, this will include your full peer review and any attached files.

**Do you want your identity to be public for this peer review?** For information about this choice, including consent withdrawal, please see our Privacy Policy .

Reviewer #1: No

Reviewer #2: No

---

## [Author Response · Author response to Decision Letter 1]

31 Jan 2025

Please see our detailed point-by-point response to reviewers comments.

---

## [Decision Letter · Decision Letter 1]

25 Mar 2025

Gene expression-based identification of prognostic markers in lung adenocarcinoma

PONE-D-24-37221R1

Dear Dr. Planck,

We’re pleased to inform you that your manuscript has been judged scientifically suitable for publication and will be formally accepted for publication once it meets all outstanding technical requirements in addition addressing the comments provided by the reviewer/s.

Kind regards,

Asmerom Tesfamariam Sengal, MD, PhD

Academic Editor

PLOS ONE

Additional Editor Comments (optional):

Reviewers' comments:

Reviewer's Responses to Questions

**Comments to the Author**

1. If the authors have adequately addressed your comments raised in a previous round of review and you feel that this manuscript is now acceptable for publication, you may indicate that here to bypass the “Comments to the Author” section, enter your conflict of interest statement in the “Confidential to Editor” section, and submit your "Accept" recommendation.

Reviewer #1: (No Response)

Reviewer #2: All comments have been addressed

2. Is the manuscript technically sound, and do the data support the conclusions?

Reviewer #1: Yes

Reviewer #2: Yes

3. Has the statistical analysis been performed appropriately and rigorously?

Reviewer #1: Yes

Reviewer #2: Yes

4. Have the authors made all data underlying the findings in their manuscript fully available?

Reviewer #1: Yes

Reviewer #2: Yes

5. Is the manuscript presented in an intelligible fashion and written in standard English?

Reviewer #1: Yes

Reviewer #2: Yes

6. Review Comments to the Author

Reviewer #1: Authors have addressed all concerns and manuscript is ready for publication.

My only suggestion would be to update Supplementary Table 2 to include some kind of division between the three models (Ki67, MCM4, TYMS), such as through use of bold row borders.

Reviewer #2: The authors of this manuscript have addressed the comments thoroughly. The research is presented well and opens new avenues for future work. I recommend this manuscript for publication and wish the authors the best of luck.

7. PLOS authors have the option to publish the peer review history of their article (what does this mean? ). If published, this will include your full peer review and any attached files.

**Do you want your identity to be public for this peer review?** For information about this choice, including consent withdrawal, please see our Privacy Policy .

Reviewer #1: No

Reviewer #2: No

---

## [Editor Report · Acceptance letter]

PONE-D-24-37221R1

PLOS ONE

Dear Dr. Planck,

I'm pleased to inform you that your manuscript has been deemed suitable for publication in PLOS ONE. Congratulations! Your manuscript is now being handed over to our production team.

Kind regards,

on behalf of

Dr. Asmerom Tesfamariam Sengal

Academic Editor

PLOS ONE